# Opportunities and Challenges of In Vitro Tissue Culture Systems in the Era of Crop Genome Editing

**DOI:** 10.3390/ijms241511920

**Published:** 2023-07-25

**Authors:** Zelalem Eshetu Bekalu, Michael Panting, Inger Bæksted Holme, Henrik Brinch-Pedersen

**Affiliations:** Department of Agroecology, Research Center Flakkebjerg, Aarhus University, DK-4200 Slagelse, Denmark; mpanting@agro.au.dk (M.P.); inger.holme@agro.au.dk (I.B.H.); hbp@agro.au.dk (H.B.-P.)

**Keywords:** crops, tissue culture, genetic transformation, new breeding technologies, genome editing, trait improvement

## Abstract

Currently, the development of genome editing (GE) tools has provided a wide platform for targeted modification of plant genomes. However, the lack of versatile DNA delivery systems for a large variety of crop species has been the main bottleneck for improving crops with beneficial traits. Currently, the generation of plants with heritable mutations induced by GE tools mostly goes through tissue culture. Unfortunately, current tissue culture systems restrict successful results to only a limited number of plant species and genotypes. In order to release the full potential of the GE tools, procedures need to be species and genotype independent. This review provides an in-depth summary and insights into the various in vitro tissue culture systems used for GE in the economically important crops barley, wheat, rice, sorghum, soybean, maize, potatoes, cassava, and millet and uncovers new opportunities and challenges of already-established tissue culture platforms for GE in the crops.

## 1. Introduction

### 1.1. Tissue Culture Systems and Progresses

There are estimated to be hundreds of thousands of plant species in the world, and only close to 74% of them have been identified [1]. Of these, humans have only fully domesticated around 250 species and moderately domesticated close to 2500 species [2,3,4]. The majority of our dietary consumption depends on around 50 plant species that offer about 95% of the world’s caloric intake [1]. Globally, the production of primary crops from 2000–2020 has increased by 52%, reaching 9.3 billion tons, an increase of 3.2 billion tons from year 2000 [5]. Sugar cane (20%, 1.9 billion tons), maize (12%, 1.2 billion tons), rice (8%, 0.8 billion tons), and wheat (8%, 0.8 billion tons) accounted for approximately half the global production of primary crops in 2020. In addition, crops such as barley, soybean, sorghum, millet, cassava, and potato contribute to human consumption, animal feed, and bioenergy. According to the Food and Agriculture Organization (FAO) (2021), maize remains the top crop in terms of production (1.21 billion tons), followed by rice (0.78 billion tons) and wheat (0.77 billion tons) (https://www.fao.org/faostat/en/#data/QCL) (accessed on 11 January 2023) (Figure 1). In terms of area harvested, wheat (220 million hectares) production covers the largest area, followed by maize (206 million hectares) and rice (165 million hectares). However, tuber crops like potato and cassava remain at the top of the chart based on yield (hg/ha). 

Despite a substantial increase in the production of primary crops, the exploding human population and the rapid reduction of arable lands due to climate change pose an existential threat to humankind [6,7]. In order to sustain the increasing trend of crop productivity under the changing climate, breeding plays a central role in the development of improved crop varieties that can tolerate the impact of climate change and the outbreak of plant diseases. Mutation breeding requires years of crossing and backcrossing to develop a trait [8]. In contrast, transgenic breeding enables the introduction of desired traits into elite varieties in a shorter time span. 

Almost all of the current plant transformation methods require regeneration from tissue culture [9,10]. Tissue culture has facilitated the transformation of several economically important monocots and dicots [11,12]. However, the process is often limited to one or two genotypes per species, and those transformable genotypes are usually characterized by poor agronomic performances [13]. The genome editing (GE) tools (defined as Sequence Directed Nucleases (SDNs) in this review) require optimized tissue culture systems that can be applied to any species and genotypes. The success of genetic transformation relies on the delivery of DNA into the cell and the regeneration of transgenic plants [9,10]. Three major DNA delivery methods are currently used for the transfer of GE tools into the appropriate explants of the major crop species, depending on the objective of the study, the crop species, and the target explant used for the transformation. To date, extensive efforts have been made to improve DNA delivery techniques via, in particular, *Agrobacterium*-mediated, biolistics, and polyethylene glycol (PEG) systems. In addition, different concentrations and ratios of auxin and cytokinin (CK) have also been examined to increase the regeneration of plants from the calli of different species and genotypes [14]. Plant regeneration relies on various factors, and the rate of regeneration often varies among species and genotypes.

The selection of an appropriate target explant is one of the main elements for successful callus induction and plant regeneration [15]. To supplement these efforts, a number of developmental regulators (DRs) have recently been characterized to improve plant regeneration. These DRs have enabled the successful genetic transformation of numerous crop species and recalcitrant inbred lines [16,17,18,19]. For most of the characterized DRs, their molecular function in plant development has been unraveled and eased the process of developing robust and genotype-independent tissue culture systems [20]. In this review, we focus on nine major crops, i.e., barley, wheat, rice, sorghum, soybean, maize, potato, cassava, and millet (Figure 1). In addition to this, we summarize all GE studies published within these crops from January 2011 until June 2022. We focus on the explants used for the GE delivery, the delivery methods, and the type of GE tool delivered (Figure 2 and Figure 3, Appendix A).

### 1.2. Genome Editing Tools Used in the Crops of the Current Review

The main GE tools used for the above-mentioned crops are SDNs. These include Zink Finger Nucleases (ZFN) developed in 2003 [21], Transcription Activator-Like Effector Nucleases (TALENs) developed in 2011 [22], and Clustered Regularly Interspersed Short Palindromic Repeats (CRISPR/Cas) developed in 2012 [23]. They can all be designed to recognize and bind to specific DNA sequences in the genome and create a double-strand break (DSB) at the targeted site. The DSB is subsequently repaired by the DNA-repair mechanisms of the plant cells, and this error-prone process can lead to a mutation at the DSB [24]. For the crops included in this review, ZFNs and TALENs have not been used in published studies since 2018. Only CRISPR/Cas-based GE tools have been reported since then (Appendix A). This is likely due to the high mutation efficiency, simplicity, and cheaper means of making a construct of CRISPR/Cas as compared to ZFNs and TALENs. Additionally, not only simplex (targeting only one specific genomic sequence) but also multiplex CRISPR/Cas constructs (targeting up to 12 genomic sequences simultaneously) can be designed [25].

#### 1.2.1. Delivery of GE Tools to the Plant Cells to Make DSBs

ZFNs and TALENs are DNA-binding proteins fused to nucleases which make a DSB at the position where the DNA-binding proteins are designed to bind. They are delivered to the cells as DNA constructs [24]. As opposed to ZFN and TALENs, CRISPR/Cas recognize the targeted DNA sequence through RNA-DNA binding. CRISPR/Cas consists of a Cas nuclease and a guide RNA [23]. In the cell, the first 20 nucleotides (frequently referred to as the protospacer) of the gRNA bind to complementary DNA sequences in the genome, and the associated Cas nuclease creates the DSB at the binding site. CRISPR/Cas systems are most frequently delivered to the cells as DNA constructs but can also be delivered as RNA constructs or as gRNA-Cas protein complexes (referred to as the ribonucleoprotein (RNP)). For GE applications, the protospacer can easily be engineered to bind to 20 complementary target nucleotide sequences in the genome. However, in order for the Cas nuclease to make the DSB, a protospacer-adjacent motif (the PAM sequence) is required at the 3′ prime end of the non-complimentary strand of the target site [23]. Currently, the most frequently used CRISPR/Cas system, named CRISPR/Cas9, is derived from *Streptococcus pyogenes* (Appendix A). In this system, the Cas nuclease is called Cas9, and the required PAM sequence is 5′-NGG, where N refers to any of the four nucleotides.

The main repair pathway of DSBs in the plant cell is through non-homologous end-joining (NHEJ), in which the broken DNA strands are simply rejoined. When the rejoining is imprecise, deletions or insertions (indels) are introduced at the site of the DSB. If the GE tool is designed to make a DSB in a coding sequence, indels can inactivate the function of the gene by changing the amino acid sequence of the open reading frame [24]. In addition, the NHEJ DSB repair system can also be used to delete or insert larger fragments at specific sites [26]. Deletion of larger fragments normally requires two DBS breaks surrounding the fragment, while insertion of larger fragments requires the simultaneous delivery of the GE tool (to make the DSB) and the fragment to be inserted. 

The other repair system of DSBs in cells is homologous directed recombination (HDR). This repair pathway requires the presence of a DNA repair template with the desired changes flanked by sequences with homology to either side of the DSB. When the DNA repair template is delivered to the cell along with the GE tool to make the DSB, the nucleotide changes will be incorporated into the chromosome at the DSB through HDR. HDR repair of DSB can be used to insert transgenes at precise chromosomal locations for promoter or gene replacement, inversions, and to make base pair substitutions [26,27]. HDR-based strategies are, however, difficult in plants because of a high preference for the NHEJ pathway, and it also requires the coordinated delivery of the GE tool and the delivery of the DNA repair template [28]. 

#### 1.2.2. Approaches to Overcome Limitations of the CRISPR/Cas9 System

Some of the key limitations of the CRISPR/Cas9 systems are the requirement for a PAM sequence in the target region, base substitutions through HDR, and off-target mutations, which are more frequently induced with CRISPR/Cas9 as compared to ZFNs and TALENs. Different approaches have been developed to overcome these limitations. 

##### Expanding the DSB Targeting Range of the CRISPR/Cas System

The targeting range of the Cas9 systems has been expanded by using mutated *S*. *pyogenes* Cas9 variants recognizing different PAM sequences [29,30,31,32,33] (Appendix A). By this strategy, one of the promising engineered Cas9 variants, called SpRY, was recently developed for human cells with a nearly PAM-less requirement [34]. SpRY, however, still needs further optimization for precise editing in plants [35]. Cas9 orthologues derived from other bacteria recognizing other PAM sequences have also been used successfully in some of the crops of the present study, including the Cas9 orthologs isolated from *S. aureus* [36,37] and *S. thermophilus* [37], where both show the same mutation frequencies as achieved with *S*. *pyogenes* Cas9. Additionally, the CRISPR/Cas12a (also called Cpf1) or the CRISPR/Cas12b systems which are variants of the CRISPR/Cas system isolated from *Franciselle novicida*, *Acidaminococcus* sp., and *Lachnospiraceeae bacterium* or *Alicyclobacillus acidoterrestis* (*Aac*), *Alicyclobacillus acidiphilus* (*Aa*), *Bacillus termoamylovorans* (*Bth)*, and *Bacillus hisashii* (*Bh*), respectively, are now used as alternatives. As opposed to the other Cas nucleases, Cas12a and Cas12b require T-rich PAM sequences that are situated at the 5′ end of the target region. Cas12a is currently used for the editing of the promoter region as it has a high affinity for T-rich PAM sequences [38,39].

##### Alternative Base Substitution Methods

Targeted base substitutions are very important mutations, as the genetic code of one amino acid can be changed to code for another amino acid. Replacement of a single amino acid in the coding region of an enzyme can often alter the activity or specificity of the protein. Thus, new techniques mediating base substitutions based on CRISPR/Cas, which omit the need for a repair template, have been developed. These include two base editing systems (BE) consisting of a gRNA and a nickase (Cas9 with one of the two nuclease domains inactivated) fused to a deaminase. The two base editing systems are named cytosine base editors (CBE), which change cytosine (C·G) to thymine (T·A) [40], and adenine base editors (ABE), which change adenine (A·T) to guanine (G·C) [41]. For other base pair substitutions, the prime editing system has been developed that enables all 12 base pair substitutions. Prime editing (PE) is a relatively new tool also based on CRISPR/Cas9 [42]. In brief, a nickase is fused to a reverse transcriptase, and a modified gRNA called pegRNA containing the desired base pair changes serves as a template for the reverse transcriptase [43]. The changes are then incorporated into the genome, making all 12 base pair substitutions possible. 

##### Reducing the Off-Targets Induced by CRISPR/Cas

The CRISPR/Cas system causes more off-targets than the ZFN and TALENs systems because CRISPR/Cas has less specific binding capacity. The protospacer can therefore bind to sequences where there are nucleotide mismatches between the protospacer and the genomic DNA sequence at positions 8 to 20 from the PAM sequence [44,45,46]. Various strategies have been developed to avoid or minimize the off-target mutations created by the CRISPR/Cas system. As the reference genome sequences are available for all crop species included in this study, the main strategy to minimize off-targets is to use software platforms for designing protospacers that are specific to the target genomic sequence. In addition, except for cassava, the pan-genome sequences of the crop species included in the current study are also available, making the protospacer design even more specific to genotypes [47]. Other tools to avoid off-targets are also available, like paired CRISPR/nCas9 nickases or paired CRISPR/dCas9-Fok1 fusions, both increasing the number of nucleotides required to recognize corresponding nucleotides in the plant genome [48]. Another strategy is to use the Cas12a or Cas12b nucleases, both possessing higher specificity than Cas9 [49,50,51]. The delivery of CRISPR/Cas tools to the cells as RNP complexes also greatly reduces the number of off-target mutations as RNP complexes degrade much faster in the cell than DNA constructs [52,53].

## 2. Major Explants Used for GE Transformation

The plant starting material is an important factor for the successful transformation of a crop species. Mostly, donor plants are grown under controlled conditions, e.g., in growth chambers. In addition, the timing of donor material collection and the type of tissue to be used are important to consider. Because there are many factors to manipulate for modification, the same transformation explant is frequently used for several species. However, not only the species but also the genotype are factors to take into account when selecting an explant for transformation, as the subsequent regeneration of plants is very species- and genotype-dependent. This factor is one of the reasons why only a few dominant genotypes and explants were used for the GE experiments in some of the species included in this study. A prime example of this is barley, where a very efficient system using immature embryos as explants has been developed but almost exclusively only works for the cultivar ‘Golden Promise’ (Figure 2A,B).

For ease of discussion, the starting materials used for the transformation of crop species included in this study are summarized into five distinct groups, i.e., immature embryos, protoplasts, callus, seed-derived tissue, and vegetative tissue (Figure 2B). Mature embryos, microspores, shoot apical meristem (SAM) derived from mature embryos, zygotes, germinating seeds, half seeds, cotyledonary nodes, or cotyledons are grouped as seed-derived, whereas tubers, leaves, stem, internodes, petiole, and meristems from the apical and lateral shoots are grouped as vegetative tissues. The two most commonly used transformation targets for GE transformation among the crops included in this study are immature embryos and calli. The tissue from which the callus is induced can differ depending on the species. 

### 2.1. Major Explants Used for the Transformation of Sexually Propagated Plants

#### 2.1.1. Embryo

A fertilized egg cell develops into an embryo via a complex developmental process called embryogenesis [54]. The zygote undergoes a series of cellular divisions and differentiations to become a mature embryo. In higher plants, embryogenesis involves many genes and embryo-specific proteins. An earlier study identified around 250 *EMB* (*EMBRYO*-*DEFECTIVE*) genes essential for normal embryo development in *Arabidopsis thaliana* [55]. The list has recently been updated to 510 *EMB* genes required for the development of an embryo [56]. This rudimentary but complex tissue is the most commonly used explant in plant tissue culture for genetic transformation. Numerous studies have reported the use of mature and immature embryos as recipient explants for the genetic transformation of monocot and dicot crops [57,58,59]. The indirect somatic embryogenesis process relies on the induction of an embryonic callus from the transformed mature or immature embryo that develops into a somatic embryo from one or more cells [60]. Like other tissue culture systems, the induction of embryogenic calli from immature or mature embryos is based on the use of different concentrations and ratios of hormones (auxins and CK) and chemical inducers, e.g., Fipexide (FPX), in the culturing media [14,57,58,59]. The hormonal requirements for callus induction and shoot and root formation from the immature and mature embryos are more species- and genotype-dependent [57]. The differential requirements of hormones and chemical inducers among species and genotypes cause variations in the time to induce calli, the potential to initiate shoots and roots, and the ability to regenerate whole plants. 

##### Immature Embryos

Immature embryos are the predominant explants used for the transformation of several crop species (Figure 2B) [61,62,63,64], and the efficiency of the system depends on various physiological and environmental factors [65]. 

The physiological maturity and size of immature embryos are the major determining factors for attaining vigorous callus induction, superior callus mass, and high regeneration rate [61,62,63,64]. For instance, immature embryos of 1.5–2 mm diameter are considered optimal for barley transformation in *Agrobacterium*-mediated transformation corresponding to 12 days post-anthesis [66]. In the case of maize, immature embryos with a 1.6–2.0 mm diameter (10–12 days after pollination) are considered ideal for maize transformation [64]. However, the size of embryos for optimal transformation differs between genotypes. In maize, the ideal immature embryo size for transformation is 1.5–1.8 mm for Hi II inbred line [67], whereas for the B104 inbred line, the diameter should be 1.8–2.0 mm [68]. A significantly lower transformation frequency has been observed for the B104 inbred line when using immature embryos of <1.8 mm diameter [64]. In wheat, an earlier study used embryos of 1.0–3.0 mm diameter for the transformation of the spring wheat cultivars Bobwhite and Fielder [69]. In that study, embryos of 2.0–2.5 mm showed better transformation frequency than embryos of <2.0 mm. Recently, an optimized and successful protocol used immature embryos of 1.0–1.5 mm of hexaploid and tetraploid wheat cultivars Fielder, Cadenza, and Kronos for transformation [70]. Therefore, with minor changes in the culturing media composition and pretreatment of the 1.0–1.5 mm immature embryos [71], the authors could obtain the transformation efficiencies of the recalcitrant Cadenza and Kronos cultivars of up to 4% and 10%, respectively [70]. The size of immature embryos optimal for transformation is more or less the same in both *Agrobacterium*-mediated and biolistics DNA delivery systems. 

In addition to the size and stage of immature embryos, the position in the spike/ear also significantly influences the transformation efficiency. Immature embryos isolated from the central spikelet give higher transformation and regeneration efficiencies than the bottom and tip of the spikelet in wheat [70]. The purity of the material from pest infestation critically affects the transformation during tissue culture. In addition, the technical expertise of personnel during pollination of outcrossing species like maize and isolation of immature embryos contribute significantly to the success in transformation frequency.

In the current review, the majority of GE studies in monocots used immature embryo explants, i.e., barley (85.7%), wheat (65.8%), maize (94%), and sorghum (87.5%) (Figure 2B, Appendix A). However, in rice, only a few studies (5.3%) used immature embryos, and in millet, no studies used immature embryo explants for GE transformation. In these species, calli induced from mature seeds are preferred as transformation targets. In the case of dicots, only a single study in soybean (1.7%) used immature embryos for transformation. 

##### Mature Embryos

Due to specific requirements for size and the maturity stage, there are multiple challenges in using immature embryos for transformation. Donor plants must be grown continuously under controlled climatic conditions to ensure a constant supply of immature embryos. In contrast, mature embryos are isolated from imbibed mature seeds that can provide an easy and consistent source of explants for transformation [72]. Compared to immature embryos, the isolation of mature embryos is easy and needs only very basic technical competencies. More importantly, a mature embryo-based transformation is often less genotype-dependent than an immature embryo-based system [15]. In addition, there are insignificant differences in the physiological maturity of embryos isolated from different seed batches. However, due to lower regeneration ability, mature embryos are used less than immature embryos for the transformation of most crops [57]. This may change in the future if embryo isolation, culturing conditions, and regeneration from mature embryos are improved. For instance, the isolation of embryos from the imbibed maize seeds with longitudinal dissection improved shoot regeneration significantly [73] or by using different combinations of auxins for improving callus induction and regeneration from the mature embryo explants, like in wheat, where regeneration frequency of immature and mature embryos from 12 wheat genotypes varied significantly when grown in the presence of 2,4-dichloro-phenoxy acetic acid (2,4-D) or Dicamba in the callus induction media [57]. For both embryo explants, the use of Dicamba induced more callus mass than 2,4-D. However, callus mass does not necessarily correlate with plant regeneration capability. In a study of the tetraploid emmer (*Triticum dicoccum*), comparing mature and immature embryos, the callus induction potential was 7% higher from immature embryos than from mature embryos. In contrast, the regeneration from immature embryos was 10–12%, lower than that of mature embryos [58]. In the current study, we found that only 2.5% of the wheat and 0.8% of the rice studies used mature embryo explants for GE applications (Appendix A). 

In wheat, the biolistic transformation of only the shoot apical meristem (SAM) of the mature embryo has been used as a successful alternative in two elite winter and one spring cultivars [74]. In the spring cultivar, an editing frequency of 1.68% was achieved, which was comparable to a previous study using immature embryos as the recipient explant. In addition, the GE of winter cultivars was possible for the first time, though the editing frequency was lower than for the spring cultivar (0.3–0.9%). In the case of rice, the GE reagents were delivered into the scutellum tissue of mature embryos of the wild rice ‘Chaling’ via *A. tumefaciens* [75]. This mature embryo-based transformation system achieved a transformation efficiency of 87–94%, and the mutation frequencies of the simplex and multiplex constructs were 60–70% and 20–30%, respectively. In both wheat and rice studies, the mutations were inherited by the next generation. 

#### 2.1.2. Microspores

Microspores (immature pollen grains) can be induced into embryogenesis when isolated at around the first pollen mitosis stage. The regenerated plants will either be haploids (and will need treatment with anti-microtubule drugs for chromosome doubling), or they will spontaneously chromosome doubling during the tissue culture [76]. Microspores can be induced to embryogenesis on specific culture media while they are still within the anthers (anther culture) or isolated (isolated microspore culture). Procedures include pretreatment of cultures with temperature, sucrose and nitrogen starvation, or osmotic stress to arrest the microspores at the right stage for embryogenesis [77]. The *A. tumefaciens* transformation procedure of isolated barley microspores has been improved through optimization of pre-culture time, medium composition, medium pH, and *A. tumefaciens* strains and density [78]. Microspores are potentially valuable explant targets for the delivery of GE tools as homozygous mutations are always obtained in the primary mutant, enabling evaluation of the impact of the mutation in primary transformants. 

Presently, successful delivery of GE tools to microspores with subsequent regeneration of mutated plants has been achieved in isolated microspore cultures of the winter barley cultivar Igri [79,80,81,82] and in anther cultures of the barley spring cultivars Compass, Flinders, Scope, Spartacus, and Golden Promise using *A. tumefaciens* as a delivery tool [83]. In wheat, GE tools have been successfully delivered to isolated microspores culture of ‘AC Nanda’ and ‘Bobwhite’ using electroporation as the delivery tool [84]. 

### 2.2. Explants Used for Transformation of Vegetatively Propagated Potato

Commercial potato cultivars are highly heterozygous clones that are vegetatively propagated. In order to maintain the genetic background of the commercial clones after transformation with GE tools, only vegetative tissue can be used for transformation. Therefore, either vegetatively derived explants are used directly as transformation targets (72.35%) or protoplasts derived from vegetative tissue are used as transformation targets (23.4%) (Figure 2B). Also, for clonally propagated species like potato, is it not possible to remove the GE construct through flowering and segregation if maintenance of the genetic background of the donor material is desired? To avoid foreign DNA in the potato crop, GE can be achieved through transient expression of the GE tools. Here, the transformed potato tissues are grown on tissue culture media without a selective agent. Regenerated mutated plants are subsequently screened for T-DNA insertions, and only mutated plants without T-DNA insertions are selected [85]. Transient PEG-mediated deliveries of DNA constructs or RNP complexes into protoplasts is often used in potato (see Section 2.4) [86]. 

As illustrated in Figure 2B and Figure 3, vegetative tissues, like stem, internode, petiole, leaf, meristem, and tubers, are rarely used as explants for GE applications of the crops included in this study except for potatoes (Appendix A). Like most other crop species in this review, potato tissue culture suffers from being genotype dependent. The variety Désirée has been used in 17 GE studies, corresponding to 40.5% of all potato studies included in this study (Figure 2A). Désirée’ is favored for GE studies due to its high response in in vitro tissue culture procedures. However, as is often the case, the favored variety has low commercial value. Beaujean and coworkers (1998) demonstrated that the callus phase of Désirée could be reduced by the addition of zeatin riboside to the regeneration media. This has led to a decrease in the level of somaclonal variation and has aided the transformation of two more commercial potato varieties [87]. 

The genotype dependency of transformation in potato is, however, not as pronounced as seen in other species like wheat and barley (Figure 2A). Especially the use of intermodal explants for *Agrobacterium* transformation has proven to be less genotype-dependent [88]. Moreover, many commercial potato cultivars can be transformed with small adjustments in transformation protocols [88]. Besides Désirée, 16 different varieties were used once for GE studies (38.1%), and 3 varieties were used in two studies (Figure 2A, Appendix A). Stem, internodes, leaf, and petiole are the most frequently used explants for potato transformation. However, some other vegetative tissues have been used in a few studies. One is meristems from the apical and lateral shoots of the potato cultivar Chicago [89]. The meristems were successfully transformed with RNP by either biolistic or vacuum infiltration. The meristem is the only explant in potato other than protoplasts that support RNP-based transformation. Regeneration of potato plants from meristem tissue is a well-established procedure described in the 1950s and is routinely used for cleaning potatoes from virus infections [89,90,91,92]. Based on the experienced procedures of meristem culture, it is likely that the utilization of potato meristems for GE can be expanded further. Another, not commonly used vegetative tissue that has been successfully used as explants for GE applications is in vitro propagated potato tubers generated from stem segments [93]. Small tubers (~1 cm) induced on in vitro grown plants were cut into fine slices for *Agrobacterium* transformation. After two days of co-cultivation, the tuber slices were washed and placed onto new regeneration media and subsequently rooting media [94]. 

### 2.3. Culture Systems Using Callus as a Direct Transformation Target

A callus has been used as a target for GE applications in most of the crop species covered in this study. A single study on wheat (1.3% of the GE examples) and sorghum (12.5% of the GE examples) and three soybean studies (5.2% of the GE examples) used a callus for GE transformation (Figure 2B). In rice, 217 GE studies (82.5%) used a callus transformation, i.e., the highest percentage among crop species where alternative methods are available. Cassava and millet GE studies have exclusively used a callus as a target explant for transformation. Although these crop species all use calli as target explants, the primary starting tissues from which the callus derives differ between species and are dependent on their way of propagation. 

Cassava is like potato, a highly heterozygous vegetatively propagated species. Friable embryonic callus (FEC), initially originating from vegetative tissue, i.e., leaf material of in vitro cultured cloned plants, is exclusively used for GE transformation with high success rates. FEC is initiated from organized embryogenic structures (EOS) using a method developed by Taylor et al. in 1996 [95]. Briefly, the somatic embryogenesis is induced from clonal leaf material by moving immature leaf lobes to MS basal medium supplemented with 20 g/L sucrose and 50 µM picloram. The FEC is induced from EOS grown on Gresshoff and Doy (GD) basal medium containing 20 g/L sucrose and 50 µM picloram. The FEC is initially sub-cultured for a second and a third cycle of growth every 3–4 weeks on GD media. This FEC tissue is used for transformation 18-20 days after the third cycle by incubating with *A. tumefaciens*. After successful transformation, plantlets can be regenerated [96]. The method described above has been further optimized for high throughput of transformation of additional cassava cultivars by changing *A. tumefaciens* cell density and adding the cephalosporin antibiotic moxalactam prior to transformation [97].

In foxtail millet (*Setaria italica* L.), an efficient *A. tumefaciens* transformation protocol using callus has been described [98]. Here, scutella-embryogenic calli derived from mature seeds are used as explant material for transformation achieving around 27% efficiency, which is high compared to previous methods [99,100]. Different media compositions were tested with the optimal setup showing the best results in different foxtail millet cultivars, indicating a strongly genotype-independent transformation method for millets [98].

The majority of the many GE studies in rice use *A. tumefaciens*-mediated transformation of calli induced on scutellum from mature seeds. A callus is used in 217 studies (82.5%), and 240 studies (80.3%) use *A. tumefaciens* (Figure 2B and Figure 3). Stable transformation and integration of *A. tumefaciens* T-DNA of monocotyledonous species was only reported a few times up until the beginning of the 1990s. In 1993, Chan et al. showed that *A. tumefaciens* was able to integrate its T-DNA into the rice genome using immature embryos [101]. In 1994, Hiei et al. presented a very efficient method for *A. tumefaciens* transformation of rice scutella-derived calli. The efficiency was similar to what was achieved using *A. tumefaciens* in dicotyledons [102], and transformants could be obtained in 3–4 months. After hygromycin selection, 50–80% of the selected calli could regenerate with a transformation frequency from 12.8 to 28.6% in three japonica rice cultivars. The time from in vitro culture to soil was around 4 months, and the method could be adapted to indica rice with only minor modifications [102,103]. Later, the transformation procedure was optimized and halved the time consumption for running the entire procedure [104]. This was achieved after adding casamino acid and proline to the media and incubating the calli at 30 °C, except during co-cultivation. Later on, further optimization was carried out by adding 2 mg/L 2.4-D and reducing the pre-culture time of seed scutellum from 2–3 weeks to 1–5 days before *Agrobacterium* transformation [105].

### 2.4. Culture Systems Using Protoplast as a Direct Transformation Target

Protoplasts are isolated from somatic plant tissues by removal of the cell wall via an enzymatic treatment where cellulase, pectinase, and xylanase are regarded as the most important enzymes. A key factor for obtaining many viable protoplasts is the osmolality in the growth media after cell wall degradation. Mannitol is used for adjusting the osmolality, and the mannitol concentration has to be optimized for each experiment [106]. After the cell wall has been removed, the protoplast plasma membrane can be transfected with, for example, DNA or RNP. The isolation of plant protoplasts was first reported more than 60 years ago using tomato root tips [107]. Nine years later, protoplast derived from tobacco mesophyll cells were infected with tobacco mosaic virus (TMV) RNA, the first introduction of nucleic acid into a plant protoplast [108]. 

Protoplasts have been used as transfection targets for GE tools in all nine crops covered by the current review except for cassava. The transfection of protoplasts is frequently used as a fast pre-screening system to test the efficacy of TALENs, Cas variants, or protospacers mutation frequency, with the aim of selecting the best target for obtaining mutated plants with standard transformation procedures [25,33,106,109,110,111,112,113,114]. 

So far, none of the wheat, barley, rice, sorghum, maize, millet, and soybean protoplasts using transiently expressed GE tools have led to the regeneration of genome-edited plants. The first transient GE expression in protoplasts of wheat was achieved in 2014 [110]. The isolation and transfection procedures took only 3 days. Protoplasts isolated from young leaves were used for PEG-mediated transfection and subsequent tissue culture [110,115]. This method has formed the basis for many subsequent fast pre-screening studies to test the efficacy of GE tools in wheat [115,116,117,118,119,120]. Similar procedures for transient protoplast transfection used as a fast pre-screening system to test the efficacy of GE tools have also been developed for maize [121] and sorghum [106]. 

In potato, it is possible to regenerate plants from transfected protoplasts. Thus, transfection of protoplasts isolated from vegetative tissue is one of the two methods used to obtain GE-mutated potatoes. The other method is via the transformation of explants derived from vegetative tissues (see Section 2.2) (Figure 2B and Figure 3). The time from potato protoplast transfection over the emergence of roots and shoots to transfer to soil was approximately 84 days [122]. In general, an important consideration when choosing protoplast as a platform for GE is the high level of genomic instability in the regenerating plants, which can lead to unforeseeable genomic changes, as investigated in potato by Fossi and co-workers [123]. 

## 3. GE Tools Delivery Techniques

### 3.1. Agrobacterium-Mediated Transformation

#### 3.1.1. *Agrobacterium tumefaciens*-Mediated Transformation

*Agrobacterium tumefaciens*-mediated transformation is the most commonly used delivery system of the GE tools and has been used in all the nine crop species covered by the current review (Figure 2, Appendix A). The plant pathogenic bacterium *A. tumefaciens* causing crown gall disease in several dicot species provides a natural bacterial system for the indirect introduction of foreign DNA into the plant cell. Briefly, the DNA delivery system depends on the presence of a Ti-plasmid in the bacterium. The Ti-plasmid contains the T-DNA region surrounded by a left and a right border of 24 bp imperfect repeats, which upon *A. tumefaciens* infection is transferred from the bacterium into the plant cell mediated by a region of virulence (*vir*) genes (including *virA*, *virB1*-*11*, *virC1*-*2*, *virD1*-*4*, *virE1*-*3*, *virF*, *virG*, *virH1*-*2* and *virK*) also present on the Ti-plasmid. 

This system can therefore be used to deliver any desired DNA sequence into the plant genome by inserting the sequence between the left and right border of the T-DNA (for more details, see reviews 113, 114). In order to easily manipulate the delivery system, a binary vector system has been developed, which is currently the most commonly used delivery system. In this system, the T-DNA surrounded by the left and the right borders has been moved to a second relatively small vector while the *vir*-genes are still located on the Ti plasmid deleted of the T-DNA region [124]. 

Plant transformation procedures using *A. tumefaciens* for gene transfer were developed in the mid-1980s for several dicot species [125]. For the dicots included in the present study, *A. tumefaciens*-mediated transformation was obtained in two potato ‘cultivars’ in 1986 [126] and in one soybean cultivar in 1988 [127]. However, the first *A. tumefaciens*-transformed cultivar of cassava was not reported until 1996 [128]. The important monocot cereals remained recalcitrant to *A. tumefaciens* transformation until the middle to late 1990s, and therefore, the particle bombardment transformation method was developed by Sanford et al. in 1987 [129]. The particle bombardment method could transform most of the important cereals (see Section 3.2). Efficient *A. transformation* systems were, however, developed in the mid to late 1990s for at least some cultivars of the most important monocot species, including rice [102], maize [130], wheat [131], and barley [132]. Later, *A. tumefaciens*-mediated transformation systems were also developed for sorghum in 2000 [133] and for millet in 2011 [134]. 

An advantage of *A. tumefaciens*-mediated transformation as compared to biolistic transformation is that fewer transgene copies (most commonly, only a single copy) are inserted into the genome. This reduces the probability of gene silencing caused by multiple transgene copies and eases the out-segregating of unwanted inserts [135]. Although *A. tumefaciens*-mediated transformation has been achieved in all crop species included in this study, the transformation efficiency varies considerably between cultivars, and some cultivars are very reluctant to *A. tumefaciens* transformation. Multiple factors are important for successful *A. tumefaciens* transformation of a specific cultivar [136]. A transformation target explant capable of regeneration is a prerequisite (as described in Section 2), but the susceptibility to *A. tumefaciens* infection is also a very important factor. There are different strains of *A. tumefaciens* differing in their *Agrobacterium* background and the Ti plasmids that they harbor [137]. The most commonly used *A. tumefaciens* strains are LBA4404, GV3101, EHA101, EHA105, AGL0, and AGL1, which differ in transformation efficiencies depending on their background and host [137,138]. Although this collection of strains can transform a wide variety of plant species, none of the strains are able to overcome the plant defense system of all recalcitrant cultivars within the different crops. The plant defense system is composed of three modules, i.e., the detection of the pathogen, the signaling that triggers plant genes involved in defense, and the plant response [138]. Genes involved in all three modules have been identified, and different strategies have been investigated to overcome the plant defense system by engineering *Agrobacterium* strains (for review, see De Saeger et al., 2021 [138]). Although none of these approaches are currently used in the crops included in this study, it might be worthwhile to investigate if higher susceptibility in recalcitrant cultivars could be achieved by the use of *A. tumefaciens* strains engineered to overcome the plant defense system. For cultivars of some of the crops, it might also be worthwhile to investigate if strains of *Agrobacterium rhizogenes* or other bacteria that can mediate DNA transfer could overcome the resistance towards infection as described below. 

#### 3.1.2. *Agrobacterium rhizogenes*-Mediated Transformation

Another important species used in indirect transformation experiments is *Agrobacterium rhizogenes*. It is closely related to *A. tumefaciens,* but unlike *A. tumefaciens* causing crown gall disease, this species causes hairy root disease [139]. The Ti plasmid, referred to as the Ri plasmid, induces root formation at the site of infection by the transfer and chromosomal integration of the T-DNA region of the Ri plasmid into the plant genome. The T-DNA region also carries, besides other genes, the *rol*-genes involved in root initiation. The roots developed on the infected explant are almost similar to the wild-type roots [139]. By transforming binary vectors into *A. rhizogenes*, the T-DNA of the binary vector, and the T-DNA of the Ri-plasmid are integrated independently into the plant genome. As a result, the T-DNA of the binary vector is expressed in the hairy roots. Since the root development on the explants is fast (only 2–6 weeks), the method is frequently used in *A. rhizogenes*-susceptible species like soybean and potato to rapidly evaluate the mutation efficacy of GE constructs prior to the generation of mutated plants via *A. tumefaciens*-mediated transformation [140,141,142]. 

Similar to *A. tumefaciens*, different *A. rhizogenes* strains have been isolated, and some of these strains have been found to expand the host range as compared to strains of *A. tumefaciens* [139]. In potato and soybean, different *A. rhizogenes*-based methods have been developed for whole plant regeneration of cultivars recalcitrant to *A. tumefaciens* transformation. In potato, it is possible to induce a callus on the hairy roots and regenerate whole plants from the generated callus. This system has recently been used in an *A. tumefaciens*-recalcitrant potato cultivar to generate mutated plants from hairy roots transformed with an *A. rhizogenes* strain containing a GE binary vector [143]. In soybean, a disarmed hyper-virulent strain of *A. rhizogenes* lacking the root-inducing T-DNA region on the Ri-plasmid was developed [139]. Disarmed *A. rhizogenes* containing GE binary vectors have been used for the generation of several mutated soybean plants [140,141,144,145] (Appendix A).

#### 3.1.3. Other Bacteria That Can Mediate T-DNA Transfer

Additional T-DNA transfer systems have been developed using other bacteria such as *Ensifer adhaerens*, *Ochrobactrum haywardense, Rhizobium etli,* and Transbacter™ collective strains [146,147,148,149,150]. These bacteria could potentially expand the plant host range. *O. haywardense* is explicitly capable of transforming soybean and has been successfully used for the simultaneous delivery of GE constructs and donor templates for HDR [149].

### 3.2. Biolistic

Biolistic, also known as particle bombardment or gene gun transformation, is a versatile physical DNA delivery method that functions independently of the cell type, species, or genotype [9]. In this delivery method, the DNA construct is coated onto particles of gold or tungsten to form a DNA/particle complex. Subsequently, the DNA/particle complex is bombarded into the target tissue by a powerful shot from a helium-pressurized gun [151]. The complex penetrates the cell membranes and delivers the DNA constructs to cells. Afterwards, the bombarded explants will be transferred into callus induction and plant regeneration media to obtain transgenic plants. Compared to the indirect DNA delivery methods, biolistic is much more efficient in the DNA transfer into the cells than the genomic integration process. As a result, it causes a higher frequency of transient expression of the transgene. Despite the lower rate of stable DNA integration, the technique enabled the stable transformation of nuclear, chloroplast, and mitochondrial genomes [152] and facilitated the transformation of some of the most recalcitrant crop genotypes (Figure 2A, Appendix A). In contrast to *Agrobacterium*-mediated transformation, biolistic transformation uses a simple expression cassette for transgene expression and facilitates the simultaneous transfer of multiple constructs. Thus, biolistic can also be used for the verification of functions of multiple genes by stacking the transgenes. 

Biolistic has been used for the delivery of GE tools in wheat (47.4% of the examples), sorghum (22.2%), soybean (13.9%), barley (11.1%), maize (7%), rice (1.9%), and potato (1.7%). On the contrary, there are no examples of the successful use of biolistics for the delivery of GE reagents into cassava and millet (Figure 3, Appendix A). For both cassava and millet, the GE reagents were, in all cases, introduced using *Agrobacterium*. Biolistic has aided the delivery of GE tools into most explant types, including immature embryos, mature embryos, calli, and vegetative tissues. 

Regardless of its versatility, biolistic-mediated transformation generates large, multi-copy, and often highly complex transgenic loci that are prone to extended recombination, instability, and silencing [9]. In a comparative study on ryegrass, biolistic transformation resulted in up to 20 transgene copies, whereas the *Agrobacterium*-mediated transformation produced a maximum of five T-DNA inserts [153]. Transgene silencing occurred in half of the transgenic plants, with transgene copies between 5 and 20. In contrast, both delivery methods showed comparable potential to generate fertile and stably expressing transgenic ryegrass lines. In a study on barley, *Agrobacterium*-mediated transformation generated twice as many transgenic plants compared to biolistic transformation [154]. Moreover, the biolistic transformation introduced more than eight copies of the transgene in 60% of the transgenic lines, whereas all transgenic lines from *Agrobacterium*-mediated transformation contained 1–3 copies of the transgene. Moreover, the majority of biolistic-derived T1 populations often had transgene silencing. In general, biolistic exhibited low transformation efficiency, low, stable transgene expression, and higher transgene silencing in barley compared to *Agrobacterium*-mediated transformation [154]. 

Optimization of the DNA delivery procedure depending on the specific species could, however, minimize the limitations and further improve the efficiency of the delivery method. For instance, in wheat, a simplified DNA/gold coating procedure in biolistic consisting of PEG and magnesium salt solutions resulted in an average transformation frequency of 9.9% as compared to the conventional biolistic protocols [155]. In addition, the collective summary of 19 independent experiments using this procedure showed a single-copy transgene integration frequency of up to 73.5% in transgenic wheat plants. This biolistic method could be used directly or with minor optimization for the transformation of other crop species and cultivars recalcitrant to *Agrobacterium*.

### 3.3. PEG-Mediated

The most common way of introducing DNA or protein into protoplast cells is via PEG. Protoplasts, DNA/protein, and often CaCl_2_ are mixed in a PEG solution. DNA-free transfection introducing RNPs (i.e., Cas protein complexed with gRNA) directly into the cells can decrease the off-target mutation rate of CRISPR/Cas and ensure that no foreign DNA is integrated into the cells [52,156]. The current review includes a total of 14 studies using PEG for GE in two different species where mutated plants have been obtained. Due to the rapid increase in the use of protoplasts for screening GE targets and for RNP transformation, the PEG-mediated transformation system requires further optimization to cover a wider range of crop species. The plasmid concentration, transfection duration, CaCl_2_ concentration, and PEG 4000 concentration are the major factors that affect the transformation efficiency of PEG-mediated transformation. Thus, optimization of one or a combination of factors for a specific species might result in higher transformation frequency. 

## 4. Major Advances in Genotype-Independent Systems for Crop GE

Transformation and regeneration of plants are highly species- and genotype-dependent, which restricts the use of GE for trait improvement in elite breeding lines. In many cases, genetic transformation can only be achieved in limited numbers of genotypes, and an effective protocol for one genotype may not be applicable to another one. The biological and genetic reasons for the variation of transformation and regeneration efficiency between genotypes are still unknown. Genetic transformations of most crops included in this review have been accomplished mainly using one major genotype (Figure 2A). For instance, close to 77% of the studies for barley transformation have used the spring barley ‘Golden promise’. And in wheat transformation, the spring wheat ’Fielder’ has been used in 32% of the studies. Due to the rapid development and increasing use of GE tools, genotype-independent transformation methods are necessary to introduce valuable traits into elite genotypes. Recently, different strategies have been developed for minimizing the genotype dependency of transformation systems. These include the use of DRs, identification of a less genotype-dependent explant in barley, and delivery systems without a tissue culture phase in maize and wheat.

### 4.1. Developmental Regulators

The recent advances in plant transformation are focused on identifying and expressing plant DRs for improving regeneration. Studies in Arabidopsis have revealed that plant regeneration is a complex physiological phenomenon whereby several DRs involve in tandem to control the process under normal and stressful conditions. Most of these DRs are activated upon the exogenous application of auxins or CKs or in response to wounding [157]. For instance, the inclusion of auxins in the callus induction media (CIM) activates BABY BOOM (BBM), whereas CKs in the shoot induction media (SIM) induce the expression of WUSCHEL (WUS). In addition to hormone-induced de novo shoot regeneration, wounding also directly activates the CKs biosynthesis pathway or through the activation of a WOUND-INDUCED DEDIFFERENTIATION (WIND)-mediated pathway [158]. The network of DRs involved in the process of callus induction and root and shoot regeneration, however, varies substantially depending on the type of target cells and tissues [157]. 

In recent years, some of the DRs identified in Arabidopsis and corresponding DRs isolated from other species have been overexpressed in various species and recalcitrant genotypes. The ectopic or constitutive expression of WUS, BBM, LEAFY COTYLEDON1 (LEC1), LEAFY COTYLEDON2 (LEC2), CUP-SHAPED COTYLEDON 1 (CUC1), ENHANCED SHOOT REGENERATION (ESR), WIND1-4, GROWTH-REGULATING FACTOR 4 (GRF4) and its cofactor GRF-INTERACTING FACTOR 1 (GIF1), and TaWOX5 have improved plant regeneration considerably. The co-overexpression of transcription factors BBM and WUS2 isolated from maize increased regeneration rates in several recalcitrant maize inbred lines [18]. Despite higher regeneration frequencies, the constitutive expression of BBM and WUS2 caused pleiotropic effects like phenotypic abnormalities and sterility. The selection of suitable maize promoters to control the temporal and spatial expression of BBM and WUS2 genes, however, alleviated the pleiotropic effects [159]. In addition to maize, the differential expression of the maize BBM and WUS2 has expedited the transformation of recalcitrant sorghum varieties [160]. The system also shortened the tissue culture time and improved the CRISPR-Cas9-mediated GE efficiency by 6.8-fold. 

LEC1 and LEC2 are B3 domain-encoding transcription factors that are central to embryo morphogenesis and cellular differentiation [161,162]. The ectopic postembryonic expression of LEC1 and LEC2 in vegetative cells induces the formation of somatic embryos in Arabidopsis. And the overexpression of wound-induced WIND1-4 genes triggers callus proliferation without the exogenous addition of plant hormones [163]. In addition, the WIND1-regulated pathway promotes CUC1-dependent shoot regeneration via upregulation of ESR1.

In a recent study, the overexpression of the wheat chimeric gene GRF4 and its cofactor GIF1 considerably improved the regeneration rate in wheat, triticale, and rice and increased the number of transformable wheat genotypes [16]. The GRF4-GIF1 system generated transgenic plants with a transformation efficiency of 27–96% in two tetraploid wheat varieties and 9–19% in two previously recalcitrant hexaploid wheat varieties. In contrast to the BBM-WUS system, transgenic wheat plants obtained from the GRF4-GIF1 are fertile with no obvious developmental defects. In addition, GRF4-GIF1 integrated CRISPR-Cas9 construct system has enabled the generation of 30 wheat lines mutated in the Q gene. More recently, the overexpression of the wheat TaWOX5 gene from the WUS family has substantially increased the transformation and regeneration of 31 common wheat cultivars without any pleiotropic effects [17]. Its overexpression also considerably improved the transformation efficiency of T. monococcum, triticale, rye, barley, and maize. The CRISPR-Cas9 constructs containing the TaWOX5 generated mutant wheat lines with an editing efficiency of 44.0% in ‘Jimai22′ cultivar and 45.5% in the transformation control cultivar ‘Fielder’. Thus, so far, CRISPR-Cas9 constructs, which include genes coding for DRs, have aided the GE efficiency in economically important crop species and genotypes.

### 4.2. Identification of a Less Genotype-Dependent Explant in Barley

As mentioned already in this review, a very efficient transformation system using immature embryos as explants has been developed for barley which, however, almost only gives rise to transformed plants when using the cultivar ‘Golden Promise’. Alternative barley transformation systems are therefore highly desired. The zygote within the in vitro cultured barley ovule is an attractive alternative transformation target. First of all, the regeneration ability of this target is high and genotype independent, and secondly, there is a direct development from zygotes into plantlets without an intervening dedifferentiated callus phase [164]. An Agrobacterium-mediated transformation procedure of the zygote within in vitro cultured barley ovules has been well-established, and even with transformation, the system seems to be less genotype-dependent [164,165,166]. The transformation system can be used with or without selection, although selection increases the transformation efficiency 3-fold [164]. Although no report on the use of GE tools for barley ovule transformation has been published yet, it seems a very promising less genotype-independent system for GE delivery in barley.

### 4.3. Delivery System without a Tissue Culture Phase in Maize and Wheat 

An alternative way of obtaining haploids in several crops is by in vivo modified pollination methods [76]. In maize and wheat, haploid plants can be obtained after pollination with maize pollen of certain cultivars. A hybrid zygote is formed, but the maize chromosomes are rapidly eliminated during the first zygotic mitosis divisions leaving no trace of the maize pollen chromosomes [167,168,169]. The haploid maize and wheat plants are then obtained via embryo rescue 14–20 days after pollination and are subsequently treated with anti-microtubule drugs for chromosome doubling to achieve double haploids in wheat plants. This method can also be used to induce haploids and mutations simultaneously when a maize or wheat cultivar is pollinated with a maize haploid inducer containing Cas9 and a sgRNA with a protospacer designed for a specific sequence in the maize or wheat genome [169,170]. Advantages of this method are that the CRISPR/Cas9 construct is not integrated into the genome of the final GE plants, that the procedure excludes a tissue culture phase, and that the method is less genotype-dependent than, for instance, the use of immature embryos as target explants for maize or wheat transformation [169,170].

### 4.4. Systemic Delivery of GE Tools to Wheat Using Viral Vectors

Plant viruses have been manipulated for the delivery of DNA constructs into plant cells [171]. Despite their limited cargo capacity and narrow host range, RNA viruses have often been exploited for viral-induced gene silencing (VIGS) [172]. Various strategies have been successfully developed that use viral vectors for the delivery of GE tools. Recently, a highly efficient heritable GE system that bypasses tissue culture has been developed in wheat using barley stripe mosaic virus (BSMV) for delivering the GE vector [173]. The heritability of mutation frequency in the next generation was 2.9–100% among the three wheat varieties, with a virus-free frequency of 53.8–100% of mutants. This viral vector system has also enabled multiplex GE and assisted the generation of Cas9-free wheat mutants by crossing BSMV-infected Cas9-transgenic wheat pollen with wild-type wheat.

## 5. Prospects and Limitations of In Vitro Tissue Culture Systems

### 5.1. Prospects

Tissue culture is essential for supporting GE applications in a number of plant species. However, due to time consumption, tediousness, and, most importantly, genotype dependency, it is also a bottleneck. To alleviate the problem related to genotype dependency, various strategies have been developed on non-crop plants that potentially could be extrapolated to crop species of major economic importance. These breakthrough technologies include the inclusion of new plant hormones and chemical inducers, use of DRs, development of viral vectors for GE tool delivery, and exploring new target explants.

In the model grass species, Brachypodium distachyon, a transformation procedure that combined mature embryos as target explant and a newly developed chemical inducer FPX increased transformation efficiency as compared to immature embryos [59]. As discussed above, the use of mature embryos in wheat alleviated the genotype dependency of transformation systems. Thus, this technology can be directly employed in the transformation of other cereal species included in this study.

Apart from this, a recent study examined the efficiency of DRs such as PLETHORA (PLT5), (WIND1), (ESR1), and WUS-BBM fusion (WUS-P2A-BBM) for the transformation of snapdragon, tomato, cabbage, and sweet pepper [19]. PLT5, WIND1, and WUS improved in planta transformation of snapdragons and tomatoes, wherein PLT5 resulted in the highest transformation efficiency in tomatoes and generated stable heritability of the transgene in snapdragons. In addition, PLT5 significantly improved shoot regeneration and transformation of two cabbage varieties and improved callus and somatic embryo formation in sweet pepper. Thus, since the molecular functions of most DRs are conserved across species, the role of PLT5 homologues and other DRs could be investigated for the regeneration and transformation of the nine crop species included in this study in detail, in order to aid GE applications [157]. Another breakthrough in the systemic use of DRs for GE applications is through de novo induction of meristems [174]. In this system, the maize WUS2 was fused with A. thaliana SHOOT MERISTEMLESS (STM) and delivered into tobacco seedlings. The simultaneous expression of both DRs and GE reagents generates transgenic and gene-edited shoots from de novo-induced meristems. This highly heritable GE and less genotype-dependent system could be used for the transformation of economically important crop species after further optimization.

Negative-strand plant RNA virus-based vectors have often been utilized for the delivery of GE tools to achieve simplex and multiplex mutations in non-crop plant species, mainly tobacco. For instance, the viral-mediated delivery of CRISPR-Cas9 simplex, multiplex, and large chromosome deletions has been possible at high mutation frequency [175]. In this system, the mutation frequency was >90%, and up to 57% of the mutants carried tetra-allelic, inheritable mutations. Another viral vector GE system utilizes the Arabidopsis thaliana Flowering Locus T (FT) RNA, an endogenous mobile RNA sequence. FT RNA enables cell-to-cell movement, and the fusion of sgRNAs to the FT might ensure entry into the shoot apical meristem and create heritable gene edits at higher frequencies [176]. In this system, Cas9-expressing plants are infected with an RNA virus containing sgRNA. Heritable mutation frequency in the progenies was from 65 to 100%, and up to 30% of the mutant progenies were edited at three different loci simultaneously.

### 5.2. Limitations

With the advent of increasing developments of new GE tools, the pace for the deployment of more genotype-independent tissue culture systems is hindered by the complexity of molecular mechanisms controlling plant regeneration. Despite the prospects of some progressive methods, they are limited to applying to a wider range of transformation platforms. For instance, viral vector systems have been shown to augment more reliable genotype-independent systems, but the use of viruses for the delivery of GE tools is often limited by their narrow host range, small cargo capacities, and, in some cases, the requirement of regenerating plants with heritable edits. In addition to viral vectors, DRs have boosted the transformation of previously un-transformable genotypes in different plant species. However, the identification and functional validation of DRs homologs in species with high ploidy levels are often complex as several copies may involve simultaneously or evolve different functions.

## 6. Conclusions

Most genetic transformation approaches are highly dependent on in vitro plant regeneration systems. At the beginning of the genome editing era, the development rested mainly on technology developed within traditional genetic engineering. At least one predominant tissue culture system was used for mainly one genotype of the crop species included in the current review. In order to facilitate more GE applications and broaden their use, additional in vitro tissue culture systems needed to be implemented. The development has largely resulted from a desire for increased speed, DNA-free use, and the ability to carry out GE, ideally in any variety within a species. Great progress has been achieved within the two first areas. However, in order to unlock the full potential of GE technology, it is still necessary that GE can be performed on any variety within the crop species. This has not yet been successful. The ovule system in barley signifying genotype independence may be a promising system that needs to be studied further. Moreover, the use of development regulators is another development that has caused some progress. The development in this area has probably just begun, and in this regard, the rapid increase of sequencing platforms and functional studies of gene functions will facilitate the identification of more DRs. So far, a very small subset of DRs has been used in the development of a more genotype-independent tissue culture system compared to the extended lists of DRs identified. Since the function of most of the DRs is conserved across species, computational analysis of the closest homologues in the crop species included in the current study will provide a more reliable transformation platform for elite genotypes.

## Figures and Tables

**Figure 1 ijms-24-11920-f001:**
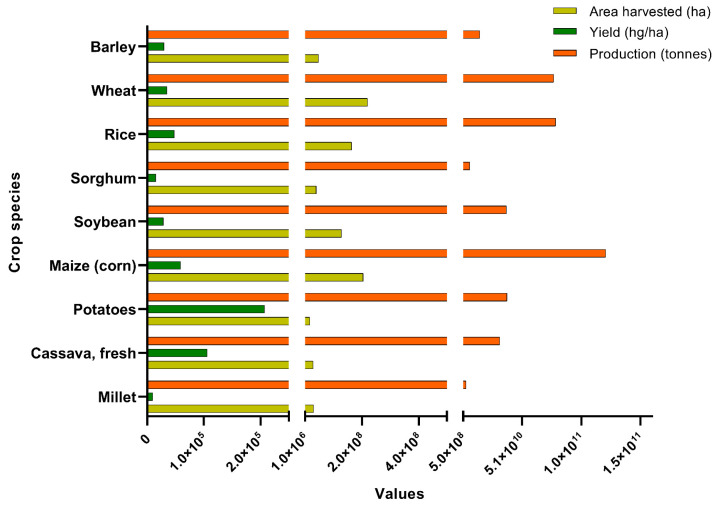
The global area harvested, yield, and production of the nine major crop species for the year 2021. The latest data was accessed from FAOSTAT on 11 January 2023 (https://www.fao.org/faostat/en/#data/QCL).

**Figure 2 ijms-24-11920-f002:**
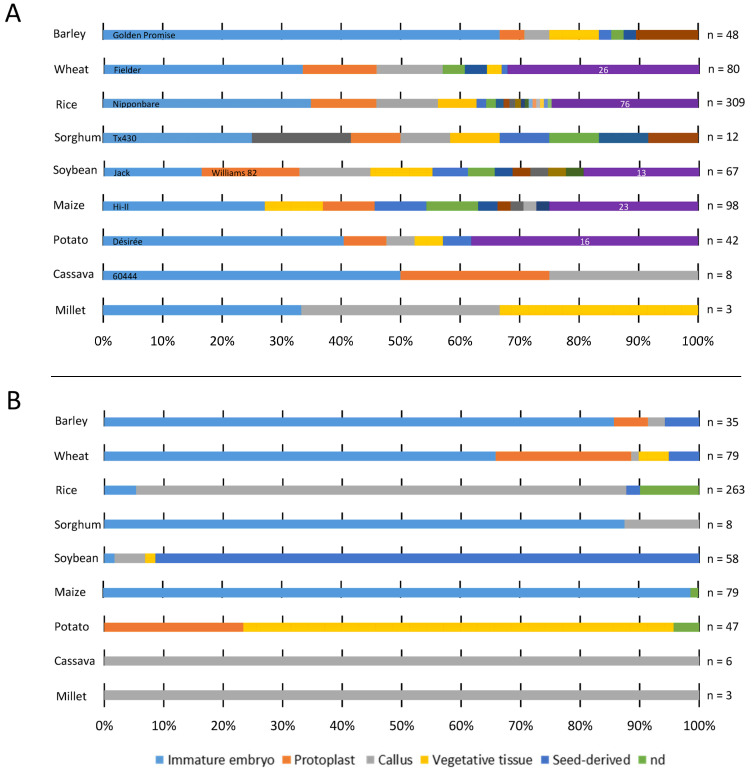
Genotypes and explants used for the GE applications among the nine crop species included in this study. (**A**) The number of genotypes used for the GE transformation of each crop species is shown as a percentage. The number of genotypes used in the studies (n) is shown next to each bar representing each crop species. Genotypes used in a few studies are pooled together and are represented in the purple bar with the number of genotypes. (**B**) The types of explants used for the GE transformation of each crop species are represented with different colored bars (legend) in percentages. The number of studies (n) is shown next to each bar.

**Figure 3 ijms-24-11920-f003:**
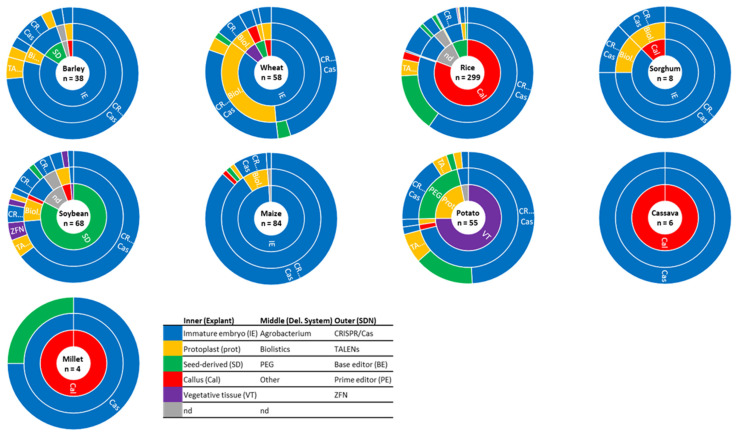
Sunburst charts depicting the combination of explants, delivery systems, and GE tools for the nine crop species included in this study. Explants (inner circle), delivery systems (middle circle), and the different GE tools (outer circle) for each species. Values are shown as percentages of the total number of GE studies (n) summarized in this study. The color coding for the different layers is shown in the legend. The figure is based on the studies included in Appendix A.

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
