# Peer review of "Opportunities and Challenges of In Vitro Tissue Culture Systems in the Era of Crop Genome Editing"

_ijms, 2023, doi:10.3390/ijms241511920_

Round 1

Reviewer 1 Report

In this manuscript “Opportunities and challenges of in vitro tissue culture systems in the era of crop genome editing”, the authors present a comprehensive, current, and well-written review of the literature providing an in-depth summary and insights into the various in vitro tissue culture systems used for GE in the economically important crops and uncovers new opportunities and challenges of already established tissue culture platforms for GE in the crops. It is well organized into appropriate subheadings, and does a commendable job summarizing recent findings and highlighting key studies. Given the importance of in vitro tissue culture in crop germplasm enhancement, the review should find a broad readership and be well received by researchers working on this field, as well as those with an interest in general crops tissue culture progresses. I would expect the review to be well cited.

For the most part, the writing is clear, concise, and easy to follow. There are a handful of minor errors, but I did not notice any misinterpretations or misrepresentations of the cited literature. The figures are nicely composed and complement the text well and readers should find them helpful.

While the main citation list may not be absolutely comprehensive, for a review of this scale this manuscript has done a good job of drawing from a range of topics and organizing them collectively. This organization in the text is complemented well by the figures.

Overall, I have no strong reservations or concerns. Below, I make a few minor comments and suggestions:

Line 11-12, the authors claimed that:” Currently, the generation of plants with heritable mutations induced by GE tools almost excollusively goes through tissue culture”.  This does not seem right, as the model plant Arabidopsis heritable mutations induced by GE tools is generated by floral dipping. I the past few years several groups reported the method by passing tissue culture in Arabidopsis, Nicotiana benthamiana, and wheat (Li et al., Mol. Plant, 11(2021)1787-1798, Ali et al., Mol. Plant, 8 (2015), pp. 1288-1291, Ellison et al., Nat. Plants, 6 (2020), pp. 620-624).

Line 34, Please provide the full name of FAO.

Line 69, additional space between “of   development”.

Paragraph two is way too long, starting from line 40- line 74. Is it better to break it down into several more paragraphs?

Line 103, I believe the PAM (NGG) should located at the 3’-prime end of the target single strand RNA, like XXXXXXXXXXXXXXXXXXXXNGG. Please check it.

Line 126, In 1.2.2 Approaches to overcome limitations of the CRISPR/Cas9 system, please add discussion on other Cas proteins, which doesn’t require at the 3’-prime end of target NGG or similar PAM, like Cpf1’s PAM is located in 5’-Prime as TTN-xxxxxxxxx target.

Paragraph starting from line 226 to line 261 is a little too long. Please consider breaking it down into several paragraphs.

Line 346, An extra space between “proven” and “to”.

Line 585, what does GE3 stands for?

Line 645, WUSCHEL can be WUS. 

Author Response

Dear Reviewer,

Thanks for the thorough review and valuable suggestion.

Please see the attachment for the responses. 

Reviewer 2 Report

The review presented by Bekal et al., is well organized and updated version of related topic. It can be accepted with some minor revisions-

1-    Overall, the article is well arranged and written which will further strengthen the knowledge of researchers/Academicians. Authors are suggested to add advantages and disadvantages of transformation techniques with future suggestions that would be more interesting for readers. And in last, overall “future prospectus” may be added with conclusion.

2-    Line No. 457, please name all Vir genes.

Author Response

(The authors gave the same response as above.)
